# Azobenzene as Antimicrobial Molecules

**DOI:** 10.3390/molecules27175643

**Published:** 2022-09-01

**Authors:** Miriam Di Martino, Lucia Sessa, Martina Di Matteo, Barbara Panunzi, Stefano Piotto, Simona Concilio

**Affiliations:** 1Department of Pharmacy, University of Salerno, Via Giovanni Paolo II, 132, 84084 Fisciano, Italy; 2Department of Agriculture, University of Napoli Federico II, 80126 Naples, Italy; 3Bionam Research Center for Biomaterials, University of Salerno, 84084 Fisciano, Italy

**Keywords:** azobenzene, antimicrobial, organometallic, azopolymers

## Abstract

Azo molecules, characterized by the presence of a -N=N- double bond, are widely used in various fields due to their sensitivity to external stimuli, ch as light. The emergence of bacterial resistance has pushed research towards designing new antimicrobial molecules that are more efficient than those currently in use. Many authors have attempted to exploit the antimicrobial activity of azobenzene and to utilize their photoisomerization for selective control of the bioactivities of antimicrobial molecules, which is necessary for antibacterial therapy. This review will provide a systematic and consequential approach to coupling azobenzene moiety with active antimicrobial molecules and drugs, including small and large organic molecules, such as peptides. A selection of significant cutting-edge articles collected in recent years has been discussed, based on the structural pattern and antimicrobial performance, focusing especially on the photoactivity of azobenzene and the design of smart materials as the most targeted and desirable application.

## 1. Introduction

Azo compounds are characterized by the presence of a -N=N- double bond to which two substituents, usually aromatic rings, are attached, making the azo compounds colourful and widely used as dyes. Today, azo dyes represent around 70% of total production and are used in printing, food, pharmaceutical and cosmetics industries [1,2,3].

They are also a well-known class of stimuli-responsive molecules [2,4,5,6]. Azobenzene can exist in two conformations, *trans* and *cis*, and their isomerization occurs upon irradiation with UV light or thermal activation. *Cis* to *trans* retro-isomerization can occur spontaneously in the dark, due to the higher stability of the *trans* isomer, or by exposure to visible light or heating. There are two main mechanisms by which isomerization can occur: rotation and inversion (Figure 1). The rotation process involves breaking the -N=N- double bond to allow free rotation around the single bond. The rotation results in the -C-N-C- dihedral angle changing, while an -N-N-C- angle remains fixed at 120°. In the inversion mechanism, an -N=N-C- angle increases to 180°, while the -C-N=N-C- dihedral angle remains fixed; this results in a transition state in which a nitrogen atom will be hybridized sp [7,8].

The two isomers differ in their absorbance profile. The absorbance spectra of the *trans* isomer are characterized by two well-separated bands in the UV-visible region: an intense band around 300–350 nm corresponding to the π → π* transition and a weaker band in the visible region, max. 450 nm, which corresponds to the n → π* transition. In the *cis* isomer, the π → π* transition is weaker, while the band corresponding to the n → π* transition is strong [8].

Azobenzene is mainly obtained through an azo-coupling reaction (Figure 2), which consists of forming the diazonium salt (-N≡N^+^) from a primary aromatic amine under acid conditions in the presence of NaNO_2_. Then, the diazonium salt undergoes a coupling reaction in basic conditions to generate the azobenzene product [9].

The isomerization of azobenzene is used as a tool to reversibly control the chemical, mechanical and/or electro-optical properties of various materials. For example, azobenzene derivatives have been used in the construction of optical switches, optical waveguides, and memory elements [4,6,10,11,12,13]. Furthermore, incorporating azobenzene in polymer chains has enabled the fabrication of photo responsive materials with macroscopic properties, that can be externally manipulated by light with high temporal and spatial control [5,14,15,16].

Recently, several studies based on the synthesis of azo compounds for their potential use in biomedical applications have been published [17,18,19]. Azo compounds can exhibit various biological activities such as antioxidant [20], antiviral [21,22], antimicrobial [23,24], anticancer [25], and antidiabetic activities [26].

Furthermore, it is reported that changes in the dipole moment and shape of azo compounds following isomerization can induce specific perturbations in biological media, such as the permeability of bacterial cell membranes [27]. On the other hand, the conformational switching of azobenzene has allowed to observe variations in the potency and efficacy of existing drugs modified with azo compounds, and to achieve the ability to modulate drug activity, thus reducing the possibility of drug resistance occurring [28,29]. The extreme ductility of azobenzene has also been exploited to functionalize polymeric materials and nanoparticles. Their incorporation has been used from time to time to develop photo responsive materials, or to control their antimicrobial activity, or to modify surface morphology.

Drug resistance is a continuously increasing problem, especially for some classes of bacteria classified as dangerous, such as *Staphylococcus aureus*, *Klebsiella pneumoniae*, *Acinetobacter baumannii*, and *Pseudomonas aeruginosa e Mycobacterium tuberculosis*. Rapidly evolving bacteria species are capable of inhibiting the pharmacological action of molecules through several types of mechanisms, such as enzymatic inactivation, modification of drug targets, or biofilm formation. The low affinity of drugs for the bacterial cell wall requires high concentrations and long-term treatments to achieve the desired benefits and performance, resulting in the occurrence of serious adverse events [18,30,31,32,33]. Therefore, it is essential to research and develop new, efficient, and non-toxic alternatives to antimicrobial drugs using known, active substances.

This review aims to present the innovations of recent years on antimicrobial compounds derived from the combination of azo compounds with biologically active molecules, polymeric matrices, and azobenzene with intrinsic antibacterial ability. In Section 2 azobenzene with antimicrobial activity are discussed, and in Section 3 azobenzene are modified and associated with biologically active molecules. Azobenzene also have interesting metal coordination capabilities, and these aspects will be described in Section 4. Finally, in Section 5, examples of the incorporation of azobenzene into polymers and nanoparticles are given.

## 2. Substituted Azobenzene Molecules with Antimicrobial Properties

Some azobenzenes exhibit intrinsic antimicrobial activity depending on the type and position of substituents on the aromatic rings. In the literature, there is a discrete number of papers reporting the synthesis and study of differently substituted azobenzene with antimicrobial and genotoxic for bacteria.

Piotto et al. [34] have reported the synthesis of five new azo compounds with different substituents on the aromatic rings (Figure 3). These compounds exhibit high antibacterial and antifungal activity against *C. albicans* (MIC_0_ = 15–30 μg/mL), *S. aureus* (MIC_0_ = 20–30 μg/mL)*, L. monocytogenes* (MIC_0_ = 25–60 μg/mL)*, S. typhimurium* (MIC_0_ > 60 μg/mL), and *P. aeruginosa* (MIC_0_ > 60 μg/mL); their antibacterial and antifungal activity is even higher than Resveratrol, a well-known natural antibiotic with a stilbene structure. 

The same authors, in a later study [35], have demonstrated that azobenzene compounds containing one or more hydroxyl groups -OH on the aromatic rings possess high antibacterial activity, particularly against *S. aureus* and *L. monocytogenes* with MICs_0_ up to 8 µg/mL.

There are several works regarding the effect on antimicrobial properties of the different substituents, and their positions on the aromatic rings of azobenzene [36,37,38,39,40]. An example was given by Ali et al. [41]. They synthesized and studied three azo compounds, in which the NO_2_ group occupies an *ortho*, *meta* or *para* position with the -N=N- double bond on one of the two aromatic rings (Figure 4).

*In vitro* antimicrobial activity was evaluated by the solid agar diffusion test, and the results showed high antimicrobial activity for the three azo dyes, superior to those of the antibiotics erythromycin and amoxicillin in the dual treatment of *S. aureus* and *C. krusei*. In particular, the molecule with the NO_2_ group at the *meta* position showed the most activity against *S. aureus* (inhibition zone of 39 mm). In comparison, the molecule in which the group is present at the *ortho* position showed the best activity against *C. krusei* (inhibition zone of 42 mm).

Another example was provided by Eriskin et al. [42]. In their work, the results showed that the different antimicrobial activity of a set of azo compounds with various substituents and in different positions is mainly due to the nature of the substituent in addition to the position (Figure 5). Molecules with electron-withdrawing groups such as chlorine (compounds **d**, **h**) showed excellent antimicrobial activity against *S. aureus*, *B. subtilis*, *K. pneumoniae*, *S. cerevisiae* and *C. albicans,* reporting MIC values up to 8.25 μg/mL. Compounds **b** and **f**, which have a nitro group in the *para* and *meta* position, respectively, were shown to be active against *S. aureus*, *B. subtilis* (MIC = 8.25 μg/mL).

However, the mechanism of action of the tested compounds remains unstudied.

Since the bacterial membrane is negatively charged, molecules with positively charged groups, such as quaternary ammonium salts, can interact with the bacterial membrane, so they are included in conventional antibacterial drugs [33]. 

Recently, in the field of modified azo molecules, amphiphilic azo compounds containing an ammonium-based cationic group, and alkyl chains with different lengths, have shown high antimicrobial activity [43]. Silico studies have shown that the antimicrobial activity is due to the interaction of the polar head of the tested molecules with the bacterial membrane, promoted by the presence of phosphatidylethanolamine (PE) in bacterial lipid membranes [44]. Among the molecules with positively charged groups that demonstrated antibacterial activity, imidazole heterocyclic derivatives play a key role in many biological systems and processes that depend on the counter ion, polar head groups, and imidazole nitrogen substituents. Similarly, the presence of fluorine atoms can increase the selectivity and lipophilicity of molecules with increased antimicrobial activity [45,46].

A recent work by Babamale et al. [47] reports the synthesis and characterization of the antibacterial properties of fluorinated and non-fluorinated azobenzene derivatives, and azo imidazole molecules with alkyl chains (R), with different lengths linked to imidazole nitrogen (Figure 6).

The compounds were tested against the gram-positive bacteria *S. aureus* and gram-negative bacteria *E. coli*, *S. enterica*, and *S. Typhimurium,* as well as against the yeasts *C. albicans* and *S. cerevisiae*, to evaluate their antimicrobial potential. Compounds **5**, **6,** and **7,** proved to be selectively active against *S. aureus* (inhibition zone of 11–14 mm), and their efficacy depends on the alkyl chain length (R) and fluorination level. In particular, the fluorinated compounds proved to be active against gram-positive bacteria, while the level of fluorination had no effect on gram-negative bacteria. Similarly, the azo imidazole molecules **11** and **12**, which have alkyl (R) chains with 16 and 18 carbon atoms, showed antibacterial activity against gram-positive species (inhibition zone of 10–11 mm), but no activity against gram-negative bacteria. The different activity against the two bacterial species is mainly due to the complexity of the gram-negative cell wall.

Salta et al. [48] described the antimicrobial activity of azo surfactants based on azobenzenes with polar groups of ammonium bromide and tobramycin (Figure 7).

The different activity of the two isomers following photoisomerization of the azobenzene moiety is also studied. The type of polar head seems to play a key role in the different activities of the two isomers. Azo compounds in which the polar head is ammonium bromide show greater antimicrobial activity of the *trans* isomer against *S. aureus* (MIC_trans_ = 1 µg/mL, MIC_cis_ = 4 µg/mL) and *E. coli* (MIC_trans_ = 8 µg/mL, MIC_cis_ = 16 µg/mL); while molecules in which the polar head is tobramycin show greater antimicrobial activity of the *cis* form against *S. aureus* (MIC_trans_16 = µg/mL, MIC_cis_ = 4 µg/mL) and *E. coli* (MIC_trans_ = 64 µg/mL, MIC_cis_ = 32 µg/mL).

The different antimicrobial activity of the two isomers is due to the different permeability effects of the membrane to the polar head.

Another study, by Velema [49], also shows the change in antimicrobial activity following photoisomerization of azobenzene derivatives. Their molecules are based on quinolones, broad-spectra antibacterial agents. The antibacterial activity of quinolones derives from binding to DNA gyrase, a key enzyme in the DNA replication process. In addition, typical quinolones used for clinical applications are characterized by the presence of piperazine and fluorine atoms on the benzene ring, which give the molecules antimicrobial activity. In Velema’s study, the quinolone moiety is linked to a photo responsive azobenzene molecule, which replaces piperazine (Figure 8). Azo-quinolone compounds with different substituents (methoxy-, methyl-, fluorine-) were synthesized and studied to investigate the effect of photoinduced isomerization of the azobenzene residue on antimicrobial properties.

The *cis* isomer showed high antimicrobial activity, while the return to the *trans* form causes a decrease in the activity of the tested molecules. The different antibacterial activity was observed in gram-negative and gram-positive bacteria, indicating that the photosensitive quinolone retains its broad-spectra activity. 

Molecule **2,** with R_2_ = Me and R_3_ = OMe, shows a significant difference in antimicrobial activity between the two isomers against *E. coli* (MIC_trans_ > 64 µg/mL, MIC_cis_ = 16 µg/mL) and *M. luteus* (MIC_trans_ = 16 µg/mL, MIC_cis_ = 2 µg/mL). 

In accordance with the previous study, further work on different antimicrobial activity following photoinduced isomerization was reported by Hu et al. [50]. In their work, the authors report the antibacterial activity of a group of carbohydrate-based surfactants with variable monosaccharide heads, including d-glucose (**AzoGlc**), D-xylose (**AzoXyl**), l-rhamnose (**AzoRha**), D-mannose (**AzoMan**), N-acetyl glucosamine (**AzoGlcNAc**), and l-arabinopyranose (**AzoAra**), linked to a hydrophobic *n*-butyl-azobenzene moiety (Figure 9).

The antibacterial activity of the surfactants was evaluated by biofilm formation against *S. aureus*, *P. aeruginosa*, and by culture broth tests against *E. coli*. The photoexcited *cis* isomers showed to be more potent against *S. aureus*, while the *trans* isomers showed higher selectivity against *E. coli*. 

Kaur et al. [25] report the synthesis of a series of diazenyl Schiff bases (Figure 10) with different donor (-CH_3_, -OCH_3_, -SCH_3_) and acceptor (-Br, -Cl, -F, -NO_2_) substituents. The antimicrobial and cytotoxic activity of these compounds was studied on cell lines. Most of the synthesized compounds showed high activity against bacteria and fungi. The structure-activity relationship revealed the importance of electron-withdrawing groups on the aromatic ring of azobenzene. In particular, the presence of bromine or chlorine on the ring of the azobenzene moiety appears to increase the antimicrobial activity (MIC up to 35 µM/mL). Di-substituted compounds on the same ring were found to be more active than mono-halogenated derivatives, and this activity was further enhanced by the presence of a third halogen.

Similarly, Mkpenie et al. [51] have shown that the presence of electron-withdrawing groups in the structure provides increased antimicrobial activity. The products, 4-((*E*)-(4-methylphenyl) diazenyl)-2-[(4-nitrophenyl)imino]methyl))phenol (**ASBn**) and 4-(((*E*)-(4-methylphenyl)diazenyl))-2-((((4-methylphenyl)imino)methyl))phenol (**ASBm**) (Figure 11) were prepared by condensation of an azo-salicylaldehyde and *p*-substituted aniline.

**ASBn** showed the best antimicrobial activity compared to **ASBm**, reporting inhibition zones of 8–12 mm and MIC = 50–250 mg/mL. Gram-positive bacteria, *S. aureus* and *S. agalactiae*, were more sensitive than gram-negative bacteria such as *P. aeruginosa* and *K. Pneumonia*. It was suggested that gram-positive bacteria are more sensitive to the synthesized compounds than gram-negative bacteria, due to the different thickness of the cell walls.

## 3. Azobenzene Ring Modification and Association with Biologically Active Compounds

One of the goals in synthesizing azo molecules associated with compounds of biological relevance, such as antibiotics and antimicrobial molecules, is to improve their biological properties. 

Organic molecules used in combination with azo moiety include heterocycles such as pyrimidines [52], indoles [53], pyrazoles [54], triazoles, benzothiazoles, and thiazoles, which have shown good antimicrobial activity. In recent years, interest in azo dyes has increased more and more due to the possibility of synthesizing derivatives of antibiotic drugs by binding with azobenzene molecules, which are more potent than originator drugs. Benzothiazole, present in many molecules of pharmaceutical interest, was combined in the work of Prakash, S. et al. [21] with four phenolic antioxidants, BHA (**4a**), floroglucinal (**4b**), 2,4-di-tert-butylphenol (**4c**), and 2,6-di-tert-butylphenol (**4d**), obtaining four new azo compounds of pharmaceutical interest (Figure 12).

Their antibacterial efficiency was tested against human pathogenic bacterial strains, such as gram-positive bacteria *B. subtilis*, *S. aureus* and gram-negative bacteria *P. aeruginosa*, *E. coli*. Their cytotoxic activity was tested against the A549 lung cancer cell line and the L929 normal healthy skin cell line; finally, antioxidant scavenging tests against free radicals and molecular docking studies were carried out. The new azo compounds showed a higher antioxidant property than the reference antioxidant BHA. Significant antibacterial effects emerged against all tested organisms. Compound **4a** showed greater antibacterial efficacy against *B. subtilis* and *S. aureus* (inhibition zones of 17 and 18 mm). Overall, the study showed good antibacterial potency for all four compounds (inhibition zones of 9–18 mm) using chloramphenicol as the standard antibiotic drug. 

The authors suggested that the lipophilicity of the compounds may have influenced the results, allowing stronger binding to the active site of the cellular enzyme. In addition, azo bonds appear to have contributed to an interaction with the enzyme’s active site via hydrogen bond formation, contributing to the inhibition of cell growth. The exact antibacterial mechanism of these new compounds has not yet been clarified, but anchoring the molecules to the bacterial cell wall seems essential. Finally, employing computational molecular docking studies, the authors showed that the new compounds have strong interaction with EGFR (epidermal growth factor receptor), an enzyme with tyrosine kinase domains, which is crucial for the development of non-small cell lung cancer (NSCLC) tumor cells.

The authors also extended their study to other biologically active molecules that incorporate the azo group in their structure, such as benzimidazole, azomethines, 2-aminothiazoles, and other benzothiazoles [55,56]. 

Another paper [57] reported the use of pyrimidine drugs and derivatives, widely used as hypnotic-sedatives, anticancer, antibacterial and anti-HIV agents, for the synthesis of new azobenzene molecules fused with 2,3-dihydro-1H pyrimidines (Figure 13).

The antibacterial efficacy of the new compounds was tested against three gram-positive bacteria species, *B. cereus*, *S. aureus,* and *S. epidermidis*, and three gram-negative bacteria species, *K. pneumonia*, *E. coli,* and *P. aeruginosa*. The new molecules (**3 a-k**) showed remarkable activity against the three gram-negative and one gram-positive bacteria (*B. cereus*). Compound **3h** was very effective against *P. aeruginosa* (inhibition zone of 16 mm), with potency similar to the reference control drug, rifamycin. Regarding the decrease in biofilm formation, compounds **3h** and **3i** were particularly effective against *P. aeruginosa* (MIC = 1.56 μg/mL and 3.13 μg/mL, respectively), also showing efficient inhibition of swimming and swarming in agar plates. Finally, anti-quorum sensing activity, which enables cell communication, was tested, demonstrating the efficacy of compounds **3h** and **3i** against the bacterial species *C. violaceum.* Again, the results were promising, causing inhibition of violacein pigment formation at 50.25% and 27.02%, respectively.

In another study [58], the authors used an antibacterial molecule of known pharmaceutical interest, sulfamethazine, to synthesize three new pharmacologically active molecules with biocidal, antitubercular, anti-inflammatory, and antioxidant activities *in vitro* by association with azobenzene molecules (Figure 14). The antibacterial activity was tested using cell culture tests, and MIC evaluation in comparison, with control drug ciprofloxacin. 

The three compounds were mildly active against *E. coli*, while they showed little efficacy against *P. aeruginosa*. Only **S3** exhibited excellent antibacterial activity against *E. faecalis* at low concentrations (up to 0.4 mg/mL). The compounds tested against the fungal species *C. albicans* were effective in the following order: **S1** > **S3** > **S2**

In the field of molecules with biological activity, Schiff’s bases are often used to prepare a wide variety of bioactive compounds; they are antibacterial, antifungal, antitumor, antiviral, and herbicidal. The azomethine bond -C=N- is responsible for the biological properties of these compounds. In a recent study by B. K. Al-Salami et al. [28], the antibacterial properties of the drug Sulfadiazine were enhanced by associating the azo and azomethine groups with the sulfamide of the drug. (Figure 15).

In the reported work, a series of new azomethine compounds were synthesized, and their antimicrobial, fungicidal, and antioxidant activity was studied. 

The antimicrobial activity was tested by the agar medium diffusion method, and the inhibition zones of the compounds were compared to those of amoxicillin. Good antibacterial activity was reported against *S. aureus* and *E. coli* for compound **1** (inhibition zones of 45 mm and 49 mm, respectively), and poor efficacy for compounds **2**–**6**. However, all compounds showed good activity against *C. albicans* and *A. niger* (inhibition zones of 15–40 mm). 

Another drug of considerable interest is aspirin, known for its anti-inflammatory and, to a small extent, antibacterial properties [59]. In a study by Z. Ngaini and N.A. Mortadza [60], the antimicrobial activity of aspirin was improved by modification with an azobenzene group and formation of analogues called azo aspirin compounds (Figure 16). The antibacterial activity of the new compounds was tested against *E. coli* and *S. aureus*.

Compounds **1a**–**d** and **3a**–**d** showed good bacteriostatic activity against both pathogenic species (MIC **1a**–**d** = 111.7–145.1 ppm against *E. coli* and MIC **1a**–**d** = 140.9–164.2 ppm against *S. aureus*; MIC **3a**–**d** = 75–94 ppm against *E. coli* and MIC **3a**–**d** = 64–89 ppm against *S. aureus*). For compound **3c** (MIC = 75 ppm against *E. coli* and MIC = 64 ppm against *S. aureus*) the bacterial growth inhibiting activity was also higher than the control drug ampicillin; while for compounds **2a**–**d**, which incorporated aspirin as such into the structure, the results appear to have been the least satisfactory. According to the authors, the mechanism of action of these new azo compounds involve hydrophobic, hydrogen bonding, and halogen interactions, that allow a higher binding strength to the active site of enzymes present on the bacterial membrane. 

The azoic bond -N=N- has been used as a binding bridge between bioactive molecules to enhance their therapeutic effects. In the Sivasankerreddy, L. et al. study, phenyl-ethyl ketone derivatives containing diazo linkage were synthesized as potential anti-infective agents [61]. 

Overall, the new diazenyl molecules proved more potent against gram-negative species, exhibiting a MIC = 6.25 mg/mL against *E. coli*.

## 4. Antimicrobial Organometallic Azo Compounds

There are several studies in the literature on the coordination of azobenzene molecules with different metal ions such as Cu (II), Zn (II), Hg (II), Co (II), Ni (II), Cr (III), and Fe (III) [2,62,63,64,65,66,67,68,69,70]. 

It is known that some azo compounds can increase their antimicrobial activity when coordinated with metal ions.

Bal et al. [63] report the antimicrobial activity of a series of water-soluble phenolic azobenzene with different substituents on aromatic rings complexed with Cu and Ni. Antibacterial activity was observed through agar well tests. All complexes, illustrated in Figure 17, showed high antibacterial activity against the bacteria *S. faecalis*, *S. aureus*, *E. cloacae*, *M. luteus,* and *E. coli,* with inhibition zones of 17–22 mm.

A new azo compound, 7-((1H-benzo [d]imidazol-2-yl) diazenyl)-5-nitroquinoline-8-ol (Figure 18), and its complexes of Cr(III), Mn(II), Co(II), Ni(II), and Cu(II), were designed and prepared in a study by Gaber et al. [64]. 

IR data suggested that the compound behaves as a tridentate ligand via the deprotonated -OH group present on the azobenzene aromatic ring in the *ortho* position, the -NH of the benzimidazole residue and an N atom of the azo group. The Cr (III) and Ni (II) complexes exhibit octahedral geometry, the Mn (II) and Cu (II) complexes square planar geometry, and the Co (II) complex has tetrahedral geometry. Antimicrobial tests have shown that complexes of Mn (II), Co (II), Ni (II), and Cu (II), are more active than the free ligand (inhibition zones of 9–16 mm) but less active than the reference drugs. As a result of complexation, the increase in antimicrobial activity is due to the positive charge of the metal ion being partially shared with the donor atoms of the ligand due to electron delocalization, with an increase in the lipophilic nature of the metal atom that favors greater permeation of the bacterial membrane. Furthermore, the Mn(II) and Co(II) complexes were found to be more active against the gram-positive bacteria *S. aureus* than against the gram-negative *E. coli*, suggesting that the antibacterial activity of these compounds is related to the structure of the bacteria’s cell membrane. The same metals have been used by Saad et al. [71] for coordination with a new azo molecule, 4-(2,4-dihydroxy-phenylazo)-*N*-thiazol-2-yl-benzenesulphonamide, (Figure 19), prepared by coupling sulphathiazole with resorcinol.

Spectroscopic data indicate a tetradentate nature of the ligand, with coordination to the two metals occurring via the deprotonated -OH group, the N atom of the thiazole ring, and the oxygen of the sulphonamide, forming a tetrahedral geometry around the metal ions. Again, antimicrobial tests, as the diffusion method in agar medium, showed higher activity of the metal complexes than the free ligand against various micro-organisms with inhibition zones of up to 11.97 mm.

Azo compounds modified with pyrazole substituents complexed with different metals are reported by Matada and Jathi [72]. In their work, the azobenzene ligand, 1,5-dimethyl-4-[(3-methyl-5-oxo-1-phenyl-4,5-dihydro-1*H*-pyrazol-4-yl)diazenyl]-2-phenyl-1,2-dihydro-3*H*-pyrazol-3-one and its metal complexes were synthesized, (Figure 20), and their antimicrobial activities against *E. coli*, *B. subtilis*, *A. niger,* and *C. albicans*, were evaluated.

Antibacterial activity test results showed an increase in activity after coordination of the ligand with metal ions. The complexes reported inhibition zones of 1.2–1.8 mm against *E. coli*, 1.4–2.0 mm against *B. subtilis*, compared to the free ligand, with inhibition zones of 1.1–1.3 mm. The metal complexes also showed antitubercular activity against *M. tuberculosis* following chelation. 

Adil A. Al-Fregi et al. [73], in a recent paper, synthesized a series of new azobenzene compounds complexed with tellurium and mercury (Figure 21).

The antibacterial activity of the tellurated and mercurated azo compounds was tested by the agar-well diffusion method against bacteria such as *S. aureus* and *E. coli*. The antibacterial activities of the tellurated azo compounds were, in some cases, comparable to or higher than those of the reference drug, with inhibition zones up to 43 mm and MICs up to 10 μg/mL. Mercurated azo compounds were found to have improved antibacterial activity than tellurated complexes.

Promising results were obtained by Kasare et al. [65]. Three new compounds based on azo-oxime (-C=N-OH) and their metal complexes with Ni (II), Cu (II), and Zn (II), (Figure 22), were synthesized, characterized, and tested, for their antibacterial and antioxidant activity. Antimicrobial efficacy was shown to be significant against gram-positive and gram-negative bacteria species, such as *S. aureus*, *B. subtilis*, *E. coli,* and *S. typhimurium*, with MICs of up to 7.81 μM/mL, using chloramphenicol as a control drug.

In a similar study [62], the antimicrobial activity of an azomethine compound capable of chelating metals such as Cu(II) and Fe(III) was investigated (Figure 23).

Antimicrobial assays were performed by the agar well diffusion method on various bacteria species, such as *S. aureus*, *E. coli*, *C. albicans*, and *A. niger*, showing inhibition zones up to 17 mm.

## 5. Azo Compounds in Antimicrobial Peptides and Polymers

The photoresponsivity of azobenzene has been used as a trigger for the formation of photo-commutable peptides, capable of changing their structure according to the isomerization state of azobenzene [74]. Lipids modified with azobenzene have received particular attention in recent years because photolipids, lipophilic molecules, can be rapidly converted from a linear to a folded form by exploiting the isomerization of the -N=N- double bond. Some authors have also used azobenzene molecules to synthesize semi-synthetic peptides, by evaluating their ability to penetrate through membranes and their behavior through the phospholipid bilayer of GUVs (Giant unilamellar vesicles) and LUVs (large unilamellar vesicles) [75]. Similarly, azobenzene molecules can be included in polymer matrices, or even be used as pendants covalently linked to polymer chains to obtain materials with various properties, such as antimicrobial. 

Infections caused by bacterial proliferation on medical devices represent a major health issue because biofilm formation is often difficult to eradicate, due to its impenetrability and strong resistance [76].

In a recent study, Kim G.C. et al. [77] studied the antimicrobial properties following isomerization of an azobenzene covalently linked to an amphiphilic peptide, consisting mainly of leucine and lysine. The study found that, following UV irradiation of the target site, *cis* azobenzene caused increased exposure of the peptide hydrophilic and hydrophobic residues, stabilizing the α-helix conformation, (Figure 24), and facilitating its penetration into the cell membrane. The authors hypothesized that the increase in peptide membrane penetration following photoisomerization is mainly due to the formation of large multimeric structures, through hydrophobic interactions between peptide molecules capable of inducing endocytosis or direct penetration.

In another study [78], azobenzene was used as a trigger to enhance the activity of a drug with a macromolecular peptide structure, gramicidin. Azobenzene appears to stabilize the β-sheet secondary structure of the peptide via intramolecular covalent bonds at the *para* and *meta* position. Gramicidin, active against *S. aureus*, contains two basic amino acids and a series of hydrophobic amino acids that enable it to bind and penetrate the anionic bacterial lipid membrane. As in the previously mentioned study, the preferred isomer here seems to be the *cis* isomer, which is more effective in stabilizing the secondary structure of the peptide, i.e., the pharmacologically active one. Six photo-commutable peptides were synthesized, and their antimicrobial activity was tested.

The *cis* **2b** isomer, represented in Figure 25, proved to be the most potent antimicrobial peptide among those synthesized with MIC = 32 μg/mL. Furthermore, replacing ornithine residues with arginine residues, as in the case of **2b**, increased membrane penetration properties and improved antimicrobial activity. 

Other authors [14] exploited the photosensitive properties of azobenzene and host-guest interactions with cyclodextrin to drive the self-assembly of AMPs (antimicrobial peptides) associated with photosensitive nanostructures, by functionalizing linear peptides with azobenzene (Figure 26a).

The synthesized peptides **P1**–**P4** have lysine side chains to provide non-specific binding sites for bacteria, while hydrophobic residues (such as valine, isoleucine, and alanine) facilitate the peptides insertion into the bacteria cell membrane; the peptide **P5**, without terminal azobenzene residues, is used as a comparison. The aim of the study was based on the recognition between azobenzene molecules and β-cyclodextrin, forming cross-linked complexes. Photoinduced isomerization was exploited to regulate the disassembly and formation of small inactive nanospheres following the reduction of host-guest interactions (Figure 26b). Antibacterial assays showed promising results of *trans*-**P1**/β-CD against *P. aeruginosa*, *S. aureus*, *B. subtilis*, and *E. coli*, with MICs of 40 μM, 15 μM, 20 μM, and 30 μM, respectively, eight times lower than the *cis*-**P1**/β-CD isomer.

In a recent paper [20], C.R. Ventura and G.R. Wiedman synthesized a new photoreactive peptide from a known peptide consisting of 26 amino acids, melittin. Melittin, known for its property of forming pores in the lipid membrane, has been widely used as an antimicrobial, anticancer, and antiviral agent. In their work, the authors interposed an azobenzene molecule in the peptide structure according to the following amino acid order: GIGAVLKVLTTGL-Z-ALISWIKRKRQQ in which Z represents the azobenzene molecule.

Lipid-loss assays were performed on lipid vesicles containing the modified peptide to test the efficacy of photomelittin, after isomerization of the azobenzene moiety and other possible uses of the liposomal content were also tested [79]. Tryptophan binding assays were also performed to test the peptide ability to bind to the membrane. UV photomelittin (*cis*-azobenzene, under UV light) proved more efficient in synthetic vesicles. In contrast, VL photomelittin (*trans*-azobenzene, under visible light) performed better in causing the rupture of human red blood cells.

Peptides modified with azobenzene have also been used to study conformational changes such as β-turn and β-sheet [80,81,82,83].

By exploiting the interdependence between light stimuli and variation in activity and pharmaceutical efficacy, in the Yeoh, Y. et al. study [84], three new antimicrobial peptides actives against *S. aureus* were synthesized by covalently binding an azobenzene molecule to the side chain or C-terminal of a starting peptide (Figure 27).

Antimicrobial assay studies revealed that azobenzene placed as the C-terminal group of the starting peptide (**3**) showed the most potent antibacterial activity, with a MIC_trans_ = 1 μg/mL. Net positive charge, hydrophobicity, azobenzene position and amphiphilicity were found to contribute to antibacterial activity by facilitating interaction with the membrane, and subsequent disrupting of the negatively charged bacterial lipid membrane.

An aminoacidic tetra-ortho-chlorobenzene was used for the modification, via covalent bonds of a well-known antimicrobial peptide, tyrocidine A, (Figure 28a), active against gram-positive pathogens [85].

Linear and cyclic analogues of tyrocidine A were synthesized, and their antibacterial activity was tested. Again, azobenzene isomerization proved to be a critical step in activating the peptide pharmacological action. Although the antimicrobial activity of tyrocidine against gram-positive species is known, some synthesized compounds, including the linear analogue shown in Figure 28b, showed surprisingly effective results against the growth of a gram-negative species, *A. baumannii*. Furthermore, after irradiation at 650 nm (*cis*-azobenzene isomer) for 40 min, the MIC of the linear compound was found to be 8 μg/mL, lower than tyrocidine A.

Azobenzene molecules were used as a bridge between two molecules of a bile acid that mimics an AMP (antimicrobial peptide). Both AMP and bile acids are composed of an extensive hydrophobic region and an extensive hydrophilic region, which are essential for anchoring to the bacterial cell membrane [86]. The authors aimed to synthesize new molecules with an antimicrobial function, not only to increase the spectra of action against bacteria that are difficult to eradicate, but also to avoid the increasingly frequent problem of drug resistance. A new pharmacologically active molecule was synthesized by using two molecules, deoxycholic, cholic or lithocholic acid (what varies is the number of hydroxyl groups in the hydrophilic region), and an azobenzene (Figure 29).

During UV irradiation, the inhibition of bacterial growth was significantly higher when the azobenzene was in the *cis* conformation, causing the molecule to assume the so-called ‘tweezer’ structure. The return to the *trans* conformation was slow enough to allow antimicrobial assays on the *cis* active form. This proved effective in inhibiting bacteria growth of *E. coli* by up to 80% at a MIC_cis_ = 100 μg/mL, while only up to 20% against the gram-positive specie, *B. megaterium*.

Chen et al. [87] report the N-terminal and C-terminal modification of **AGP** tripeptides with a hydrophobic n-butyl-azobenzene tail, and a hydrophilic N-methyl-D-glucamine residue, respectively (Figure 30).

The introduction of an azobenzene moiety was exploited to give the azobenzene-glycopeptide assemblies a reactivity to light stimuli and host-guest interactions with β-cyclodextrin (β-CD) and adamantane (Ada). The role of adamantane is to compete with the azobenzene molecule in the interaction with β-cyclodextrin.

**AGPs** showed the ability to assemble and disassemble reversibly upon isomerization of the azobenzene molecule, or through host-guest chemical interactions. **AGP** assemblies and hydrogels were prepared, on which rheological tests were performed, and antibiofilm activity on *S. aureus* was studied. The **AGP** assemblies showed anti-biofilm activity (removal efficiency of ∼90%) based on the reversible disassembly-assembly process; the antibiofilm activity could be reversibly deactivated and activated. The authors hypothesized that since the bacterial cell wall usually consists of a high number of glycosyl groups in the peptidoglycan, bacteria have a strong ability to form hydrogen bonds with similar molecular structures; therefore, the self-assembled **AGP** nanostructures have a strong affinity for the bacterial cell wall. 

In another study, the same authors used an azobenzene to form amphiphilic tripeptides (**PA**) [88]. Again, the strategy was to incorporate azobenzene into peptides, in this case, amphiphilic peptides, and to calibrate the antibacterial activity by non-covalent binding with β-cyclodextrin (β-CD), subsequently dissociated from adamantane (Ada) (Figure 31).

The authors tested antibacterial and antibiofilm properties by photoirradiation and host-guest interactions. Upon complexation with β-cyclodextrin, there was a reduction in antibacterial activity due to the interruption of their amphiphilic nature. Similarly, following photoirradiation, the molecular geometry of the *cis*-isomer restricted the close packing of **PA** molecules, thus causing their disassembly and loss of antibacterial activity.

Photo controllable antibacterial activity was tested against *E. coli*, *S. aureus*, and *P. aeruginosa*. Both compounds **PA1** and **PA2** showed an IC_50_ (50% inhibitory concentration) < 0, 2 mM against *E. coli* and *S. aureus*, while no inhibition was observed against *P. aeruginosa*. 

In a work by Diguet, A. et al. [89], the authors used an azobenzene molecule with a surfactant structure called **azoTAB** (trimethylammonium bromide) for the rupture of GUVs. The **azoTAB**, following light-induced isomerization, enabled the rupture of vesicles formed by dioleoylphosphatidylcholine (DOPC), dipalmitoylphosphatidylcholine (DPPC), and different amounts of cholesterol (Figure 32a,b).

Based on these results, other studies have been conducted using azobenzene for the alternating release of lipid vesicle content [90,91,92,93]. 

Biofilms can form on biotic and abiotic substrates, from tooth surfaces within the oral cavity to biomedical devices. Therefore, antibacterial strategies that prevent the formation and proliferation of biofilms, by controlling the material properties of the bacteria-substrate interface, have attracted much interest. For this purpose, PLA-based polymer films doped with azobenzene with antimicrobial properties were developed by Concilio et al. [94]. The concentration of up to 0.01% (w/w) of the azo compounds, synthesized by the same authors [34], (Figure 33), enabled the realization of antimicrobial and transparent films that retained the properties of pure PLA matrices, such as glass transition temperature, flexibility, and amorphous nature [95,96]. In addition, the films showed antimicrobial activity and the ability to inhibit biofilm formation of *S. aureus* and *C. albicans*. 

In recent years, polymers functionalized with azobenzene have emerged as a class of biomaterials defined as intelligent because they can use light energy to induce macroscopic deformations, or design materials where intrinsic antibacterial activity is required, exploiting the isomerization process [97,98,99].

In a recent work of Zheng et al. [100], azobenzene is used in an antibacterial and photo responsive polymeric material based on host-guest interactions. The material, obtained by surface modification, consists of an azo-functionalized antibacterial polymer, poly((2-(methacryloxy)ethyl)) trimethyl ammonium chloride (Azo-PMETAC), an antifouling polymer, and an azo-modified poly(sulfobetaine methacrylate) (Azo-PSBMA) (Figure 34a). The two polymers were anchored to a surface of poly(2-hydroxyethyl methacrylate) (PHEMA), containing -cyclodextrin groups (Figure 34b). The new material proved to be an excellent antibacterial agent, not only in inhibiting bacterial growth, but also in causing the polymer structure to break down following isomerization, causing the release of all bacteria that had grown on the surface, thus preventing the formation of biofilms. In addition, the authors demonstrated that following repeated cycles of irradiation with visible light at 450 nm, the previously disassembled polymer could be regenerated, restoring its antibacterial properties. The inhibition of bacterial growth was tested on *S. aureus* and *E. coli* showing a low bacterial cell growth of ∼6.25 × 10^5^ cells per cm^2^ and high efficiency of approximately 88%. 

Mori et al. [101] used glass acrylic resins functionalized with a hydroxylated azobenzene (OH-AAZO), (Figure 35a), to inhibit the growth and bacterial proliferation of pathogens such as *S. mutans*, *S. oralis*, *A. actinomycetemcomitans*, *S. aureus,* and *E. coli*. By covalently linking OH-AAZO to the acrylic substrate, a non-pathogen-specific antibiofilm coating was developed that utilizes light-induced mechanical movement to destroy biofilms. The antibacterial activity of OH-AAZO was tested against bacterial strains such as the commensal bacteria *S. oralis*, a gram-positive species capable of inducing severe infections in immunocompromised or chronically diseased individuals, and against *S. mutans*. Growth and proliferation tests showed good activity with MIC = 25 mg/mL and complete sterility at 50 mg/mL. No inhibition was observed against bacterial species such as *A. actinomycetemcomitans*, *S. aureus*, and *E. coli* (Figure 35b).

Devatha P. Nair et al. [102] exploited the photoresponsivity of azobenzene as a means of inhibiting biofilm growth through a process of photo-fluidification, i.e., softening the surface of a glassy material.

The glassy material, consisting of PMMA/MMA/TEGDMA (poly(methyl methacrylate)/methyl methacrylate/triethylene glycol dimethacrylate), was covalently linked to the surface using thermal cross-linking to a photo responsive azobenzene molecule (AZO) (Figure 36).

Efficacy was tested against *P. aeruginosa* biofilm preformed on the photosensitive polymer substrate, and following successive cycles of *trans*-*cis*-*trans* isomerization of the azobenzene, to promote its detachment. The biofilms on the surface of the modified AZO material and the control substrate (without AZO) were exposed to the light of a dental lamp to start the fluidification effect, and subsequently washed with PBS to remove the detached bacteria from the biofilm, showing effective inhibition. 

Inhibition of biofilm growth was also tested by incorporation of azobenzene-functionalized nanogels into BisGMA/HEMA/ethanol (bisphenol A-glycidylmethacrylate/hydroxyethyl methacrylate/ethanol) dental adhesive resins, hereinafter called B/H/E (Figure 37) [103]. 

The authors investigated the incorporation of nanogels, having a hydrodynamic radius < 2 nm and a molecular weight of 12,000 Da, functionalized with azobenzene within dental adhesive resins. The aim was to increase the hydrophobicity of the biomaterial, while maintaining its mechanical and adhesive performance. In particular, the new resin was tested to reduce the presence of cariogenic bacteria, such as *S. mutans,* on the modified resin substrates. The antibacterial biofilm test showed that the presence of *S. mutans* on the substrates was significantly reduced, compared to the control; furthermore, the azobenzene/B/H/E functionalized nanogels showed a ∼66% reduction in colonies on the surface of the light-cured substrates, compared to the azobenzene-free B/H/E control. 

Other uses of azo compounds in polymer matrices have found interest in food packaging. In the study of Marturano et al. [104], a photosensitive azobenzene was exploited for the release of a well-known antibacterial active ingredient, thyme essential oil, by coating commercial packaging films, such as polyethylene (PE) and polylactic acid (PLA), with photoreactive nanocapsules (NC) containing the active agent (Figure 38). 

UV-Vis and FTIR characterization of the films, to evaluate the response to UV irradiation, showed that the immobilization of NCs on the polymer surface did not hinder the *trans*-to-*cis* isomerization of azobenzene. Furthermore, a 15 min UV exposure led to an 8-fold increase in the concentration of thyme oil in the headspace of the polymer coatings after 24 h, thus confirming the efficiency of the light-triggered system.

## 6. Conclusions

The emergence of bacterial strains resistant to antibiotic drugs is a severe threat in the prevention of infections and resulting diseases. Current drugs are proving less and less efficient, and new, alternative antimicrobial therapies are needed. In this context, the combination of different classes of bioactive molecules allows us to obtain new compounds with an enhanced effect, being less susceptible to bacterial resistance. The mechanisms by which azobenzenes or their derivatives exert their activity remain unclear and probably act on different targets. For this purpose, to facilitate the future work of many researchers, and to encourage the design of azobenzene-based antibacterial compounds, we have prepared a summary table, (Appendix A, See Appendix A), with the relationships between chemical substitutions on azobenzene rings and the effect on antibacterial activity. Where the target is a protein receptor (ATP synthase is an enzyme that has often been suggested), it is easy to imagine how only one of the two forms (*cis* or *trans*) can fit the geometry of the binding pocket. The change in geometry can be radically compromised when azobenzenes are coordinated to a metal ion or covalently bound in a polymer. In a polymer matrix, any transitions can lead to drastic changes in the morphology of the material. Where the target is a cell membrane, at concentrations not so high as to lead to its complete destruction, the change in *trans*-*cis* geometry can increase membrane fluidity and ion diffusivity. In recent years, numerous azo compounds with antibacterial properties have been synthesized and studied, in combination with molecules with known activity. In addition, using the photosensitivity of azobenzene allows a spatio-temporal control of the molecules antimicrobial activity by using light. This could be particularly useful in the treatment of bacterial infections by reducing the side effects of antibiotics.

Similarly, azobenzene-modified polymers with antimicrobial activity are finding use in the medical field, in the prevention of biofilm formation on substrates and medical devices, and in the design of smart materials with reversible properties.

## Figures and Tables

**Figure 1 molecules-27-05643-f001:**
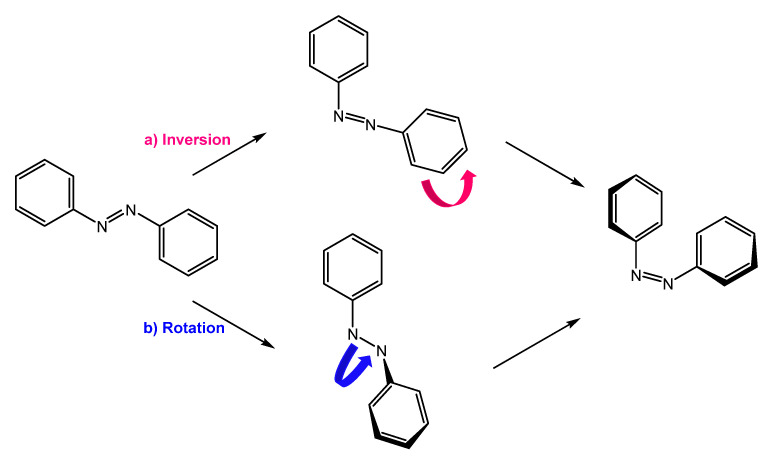
Isomerization by (**a**) inversion and (**b**) rotation mechanism.

**Figure 2 molecules-27-05643-f002:**
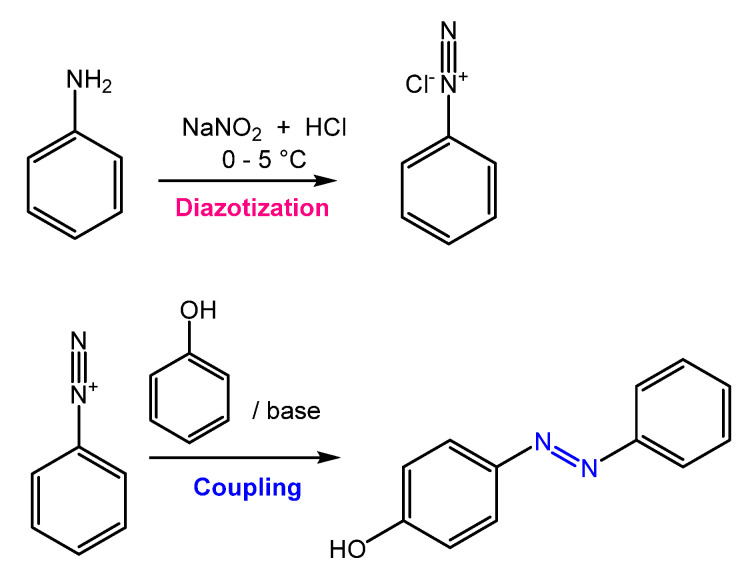
General scheme of the azo coupling reaction.

**Figure 3 molecules-27-05643-f003:**
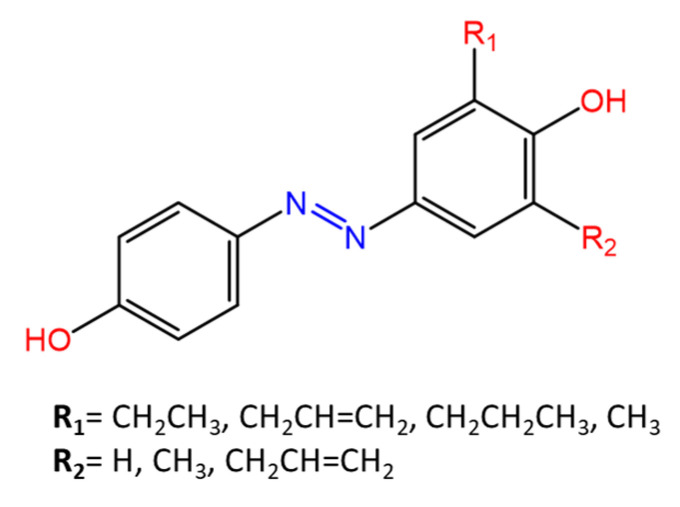
Structure of antimicrobial azo compounds [34].

**Figure 4 molecules-27-05643-f004:**
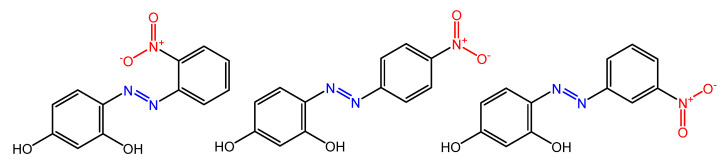
Nitro-substituted azobenzenes by Ali et al. [41].

**Figure 5 molecules-27-05643-f005:**
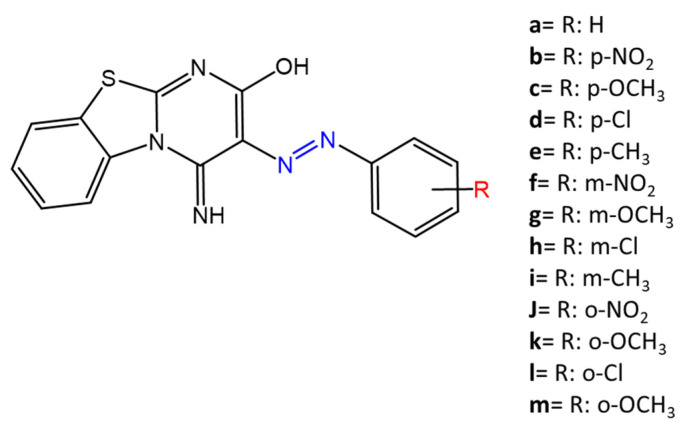
Azobenzene differently substituted by Erişkin et al. [42].

**Figure 6 molecules-27-05643-f006:**
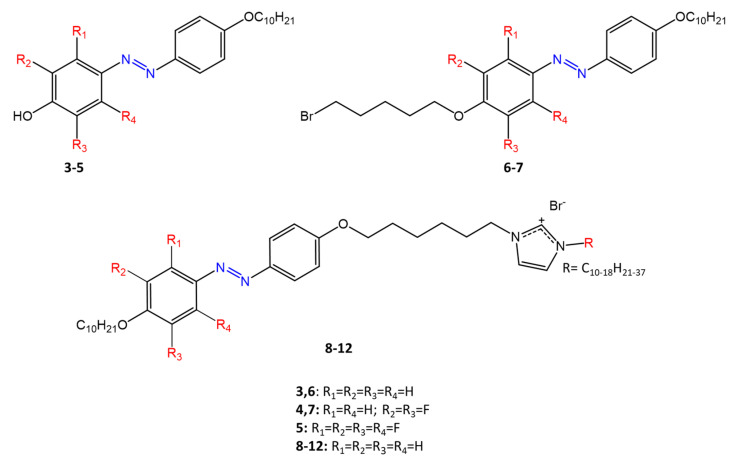
Fluorinated and non-fluorinated azobenzene derivatives and azo imidazole molecules [47].

**Figure 7 molecules-27-05643-f007:**
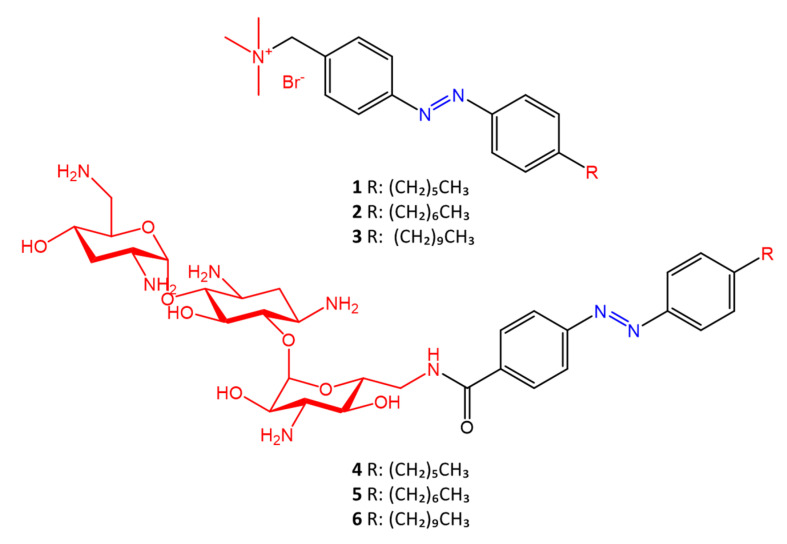
Azobenzene compounds with ammonium and tobramycin polar heads (in red) from Salta et al. [48].

**Figure 8 molecules-27-05643-f008:**
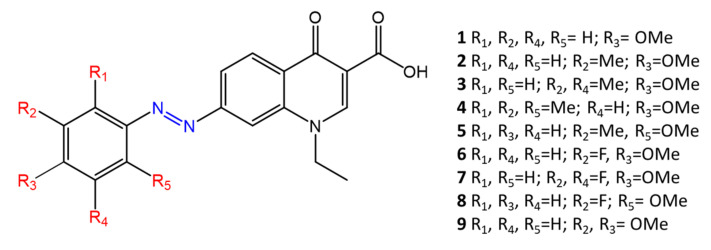
Chemical structure of azoquinolones from Velema [49].

**Figure 9 molecules-27-05643-f009:**
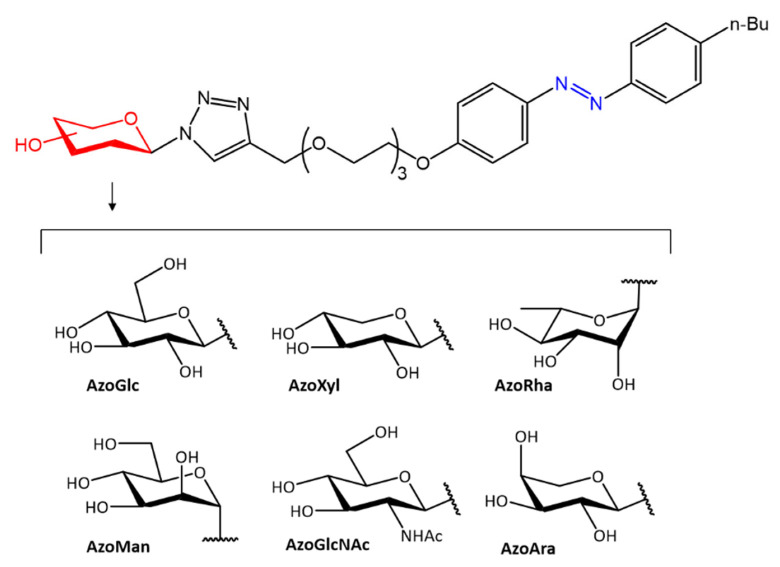
Photoreactive carbohydrate-based surfactants [50].

**Figure 10 molecules-27-05643-f010:**
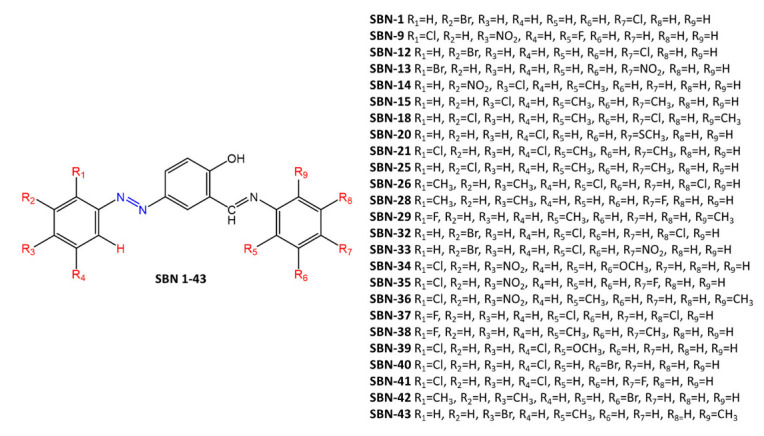
Chemical structure of Schiff-base diazenyl compounds [25].

**Figure 11 molecules-27-05643-f011:**
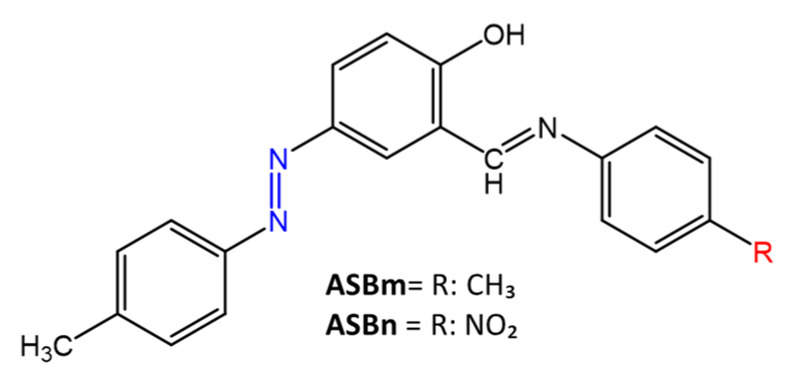
Chemical structure of azo Schiff bases [51].

**Figure 12 molecules-27-05643-f012:**
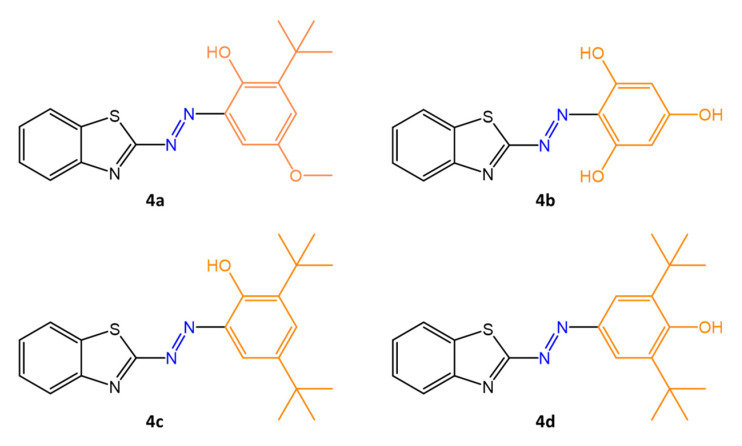
Chemical structure of azobenzothiazole compounds with different antioxidant molecules [21].

**Figure 13 molecules-27-05643-f013:**
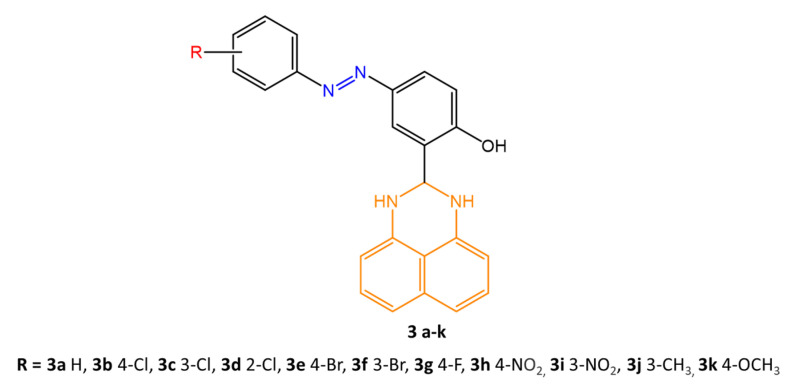
Chemical structure of novel (aryldiazenyl)-2-(2,3-dihydro-1H-perimidin-2-yl)phenol derivatives [57].

**Figure 14 molecules-27-05643-f014:**
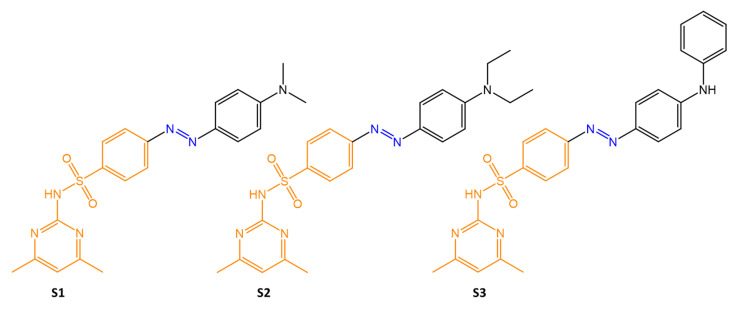
Chemical structures of *N*,*N*-dimethyl aniline derivates based azo dyes [58].

**Figure 15 molecules-27-05643-f015:**
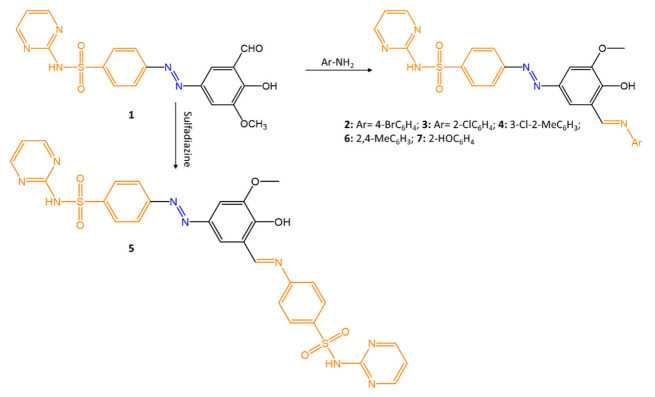
Sulfa drug containing azo-azomethine group [28].

**Figure 16 molecules-27-05643-f016:**
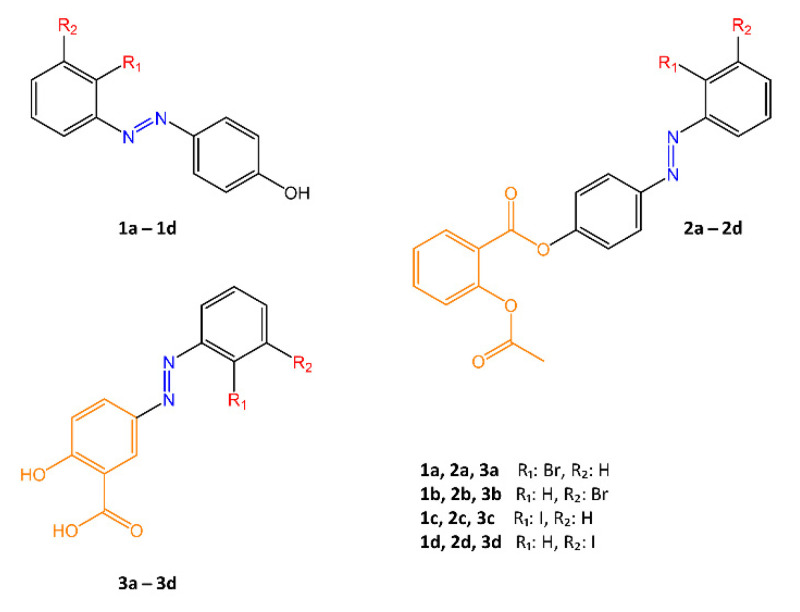
Chemical structures of halogenated azo aspirin analogues [60].

**Figure 17 molecules-27-05643-f017:**
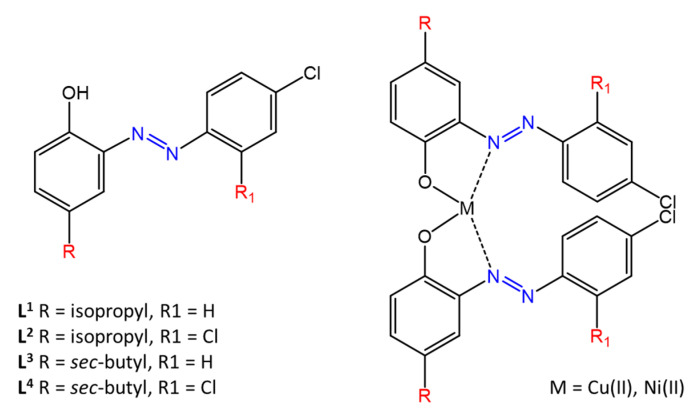
Structures of phenolic azo dyes and their Ni(II) and Cu(II) complexes [63].

**Figure 18 molecules-27-05643-f018:**
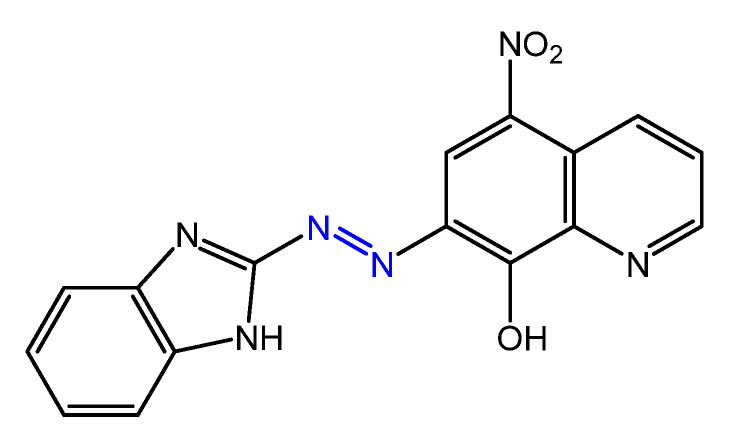
Structure of 7-((1H-benzo [d]imidazol-2-il) diazenil)-5-nitrochinolina-8-olo [64].

**Figure 19 molecules-27-05643-f019:**
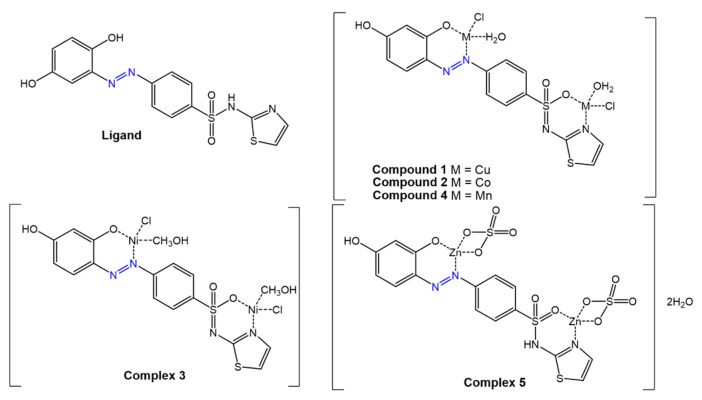
Structure of the sulfathiazolyl azo-resorcinol ligand and its metal complexes [71].

**Figure 20 molecules-27-05643-f020:**
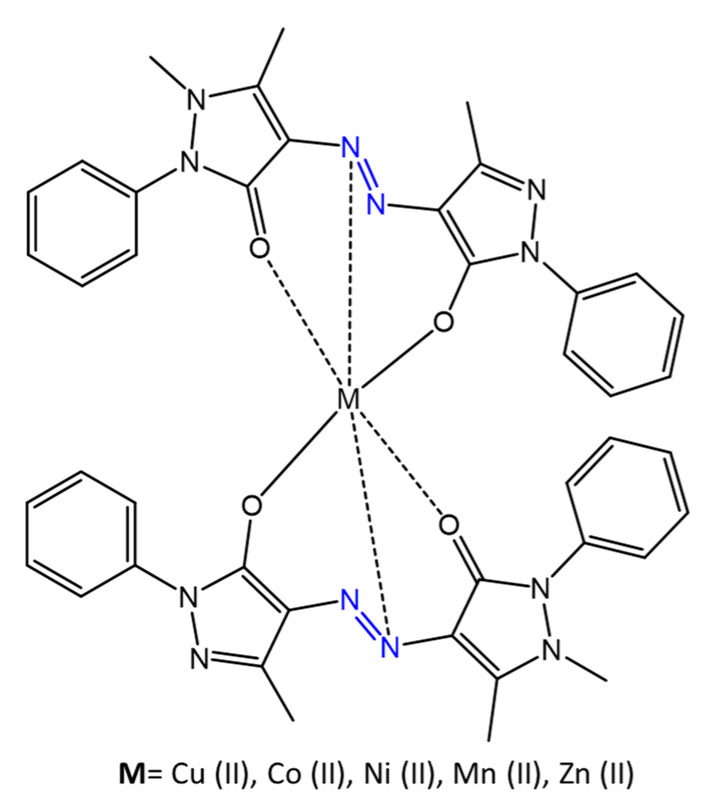
Pyrazole-based azo-metal (II) complexes [72].

**Figure 21 molecules-27-05643-f021:**
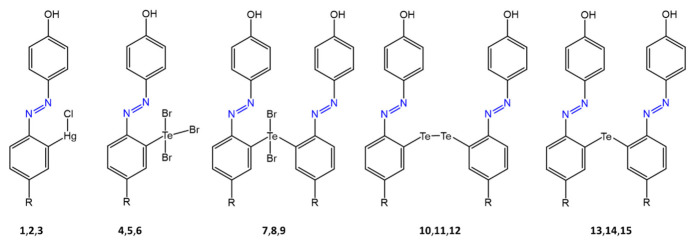
Chemical structures of tellurated and mercurated azo compounds [73].

**Figure 22 molecules-27-05643-f022:**
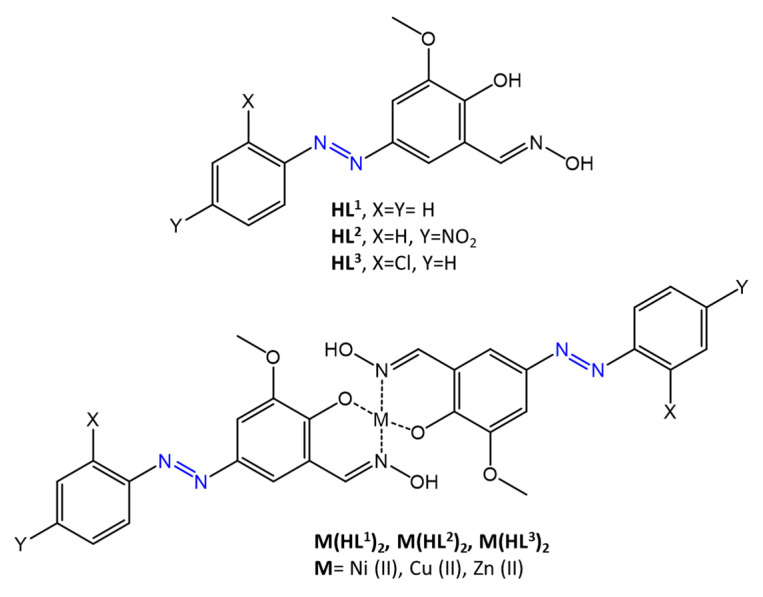
Chemical structures of azo-oxime ligands and their complexes [65].

**Figure 23 molecules-27-05643-f023:**
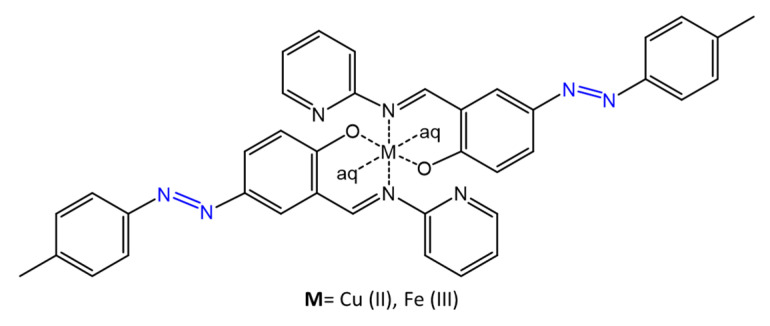
Chemical structures of azomethine complexes [62].

**Figure 24 molecules-27-05643-f024:**
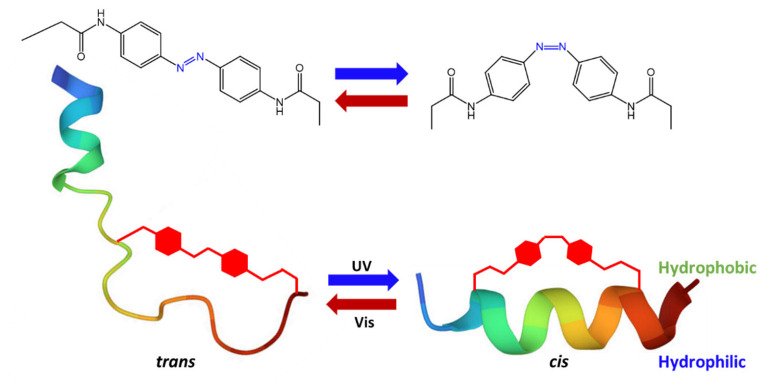
Scheme of photoswitching α-helical structure [77].

**Figure 25 molecules-27-05643-f025:**
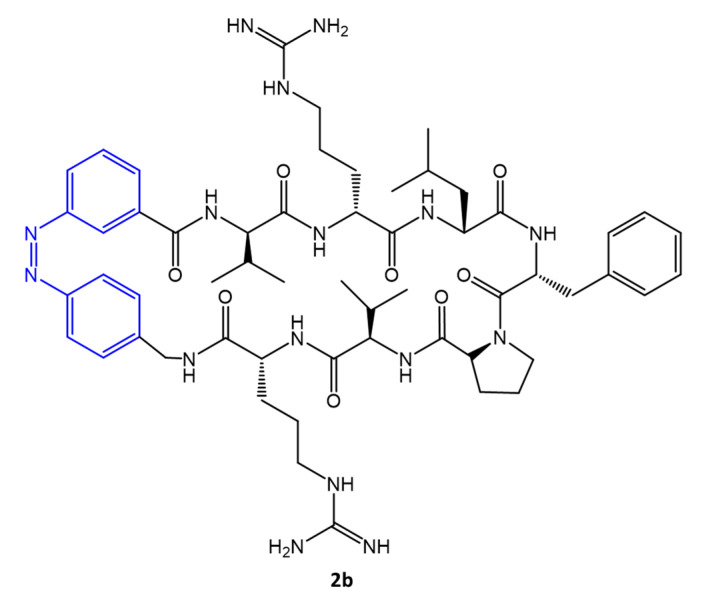
Structure of 2b mimetic peptide of gramicidin [78].

**Figure 26 molecules-27-05643-f026:**
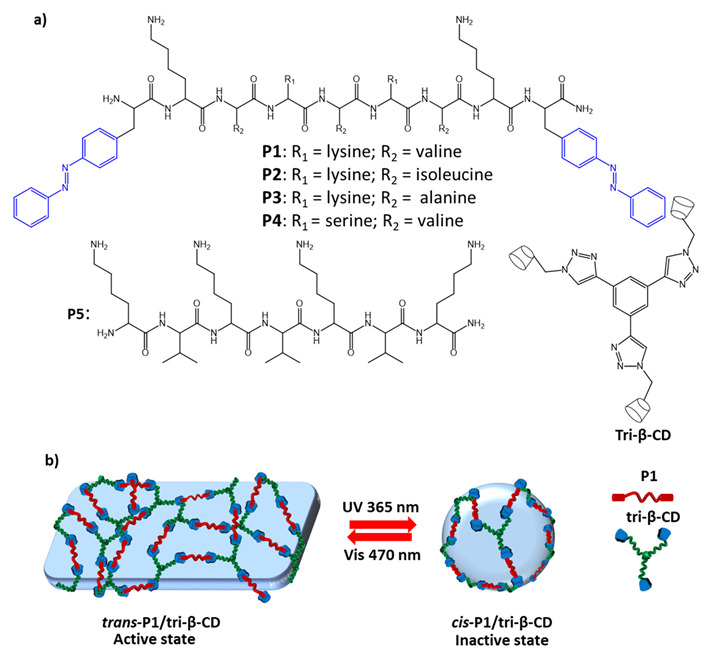
(**a**) chemical structures of peptides P1-P5, and tri-β-CD; (**b**) scheme of photoresponsive dynamic self-assembly [14].

**Figure 27 molecules-27-05643-f027:**
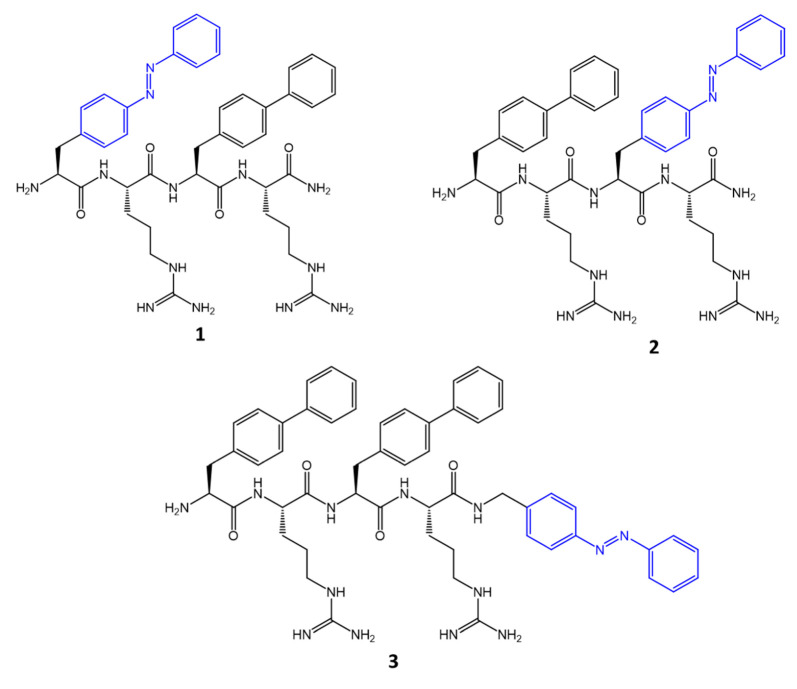
Structures of short photoswitchable tetrapeptides [84].

**Figure 28 molecules-27-05643-f028:**
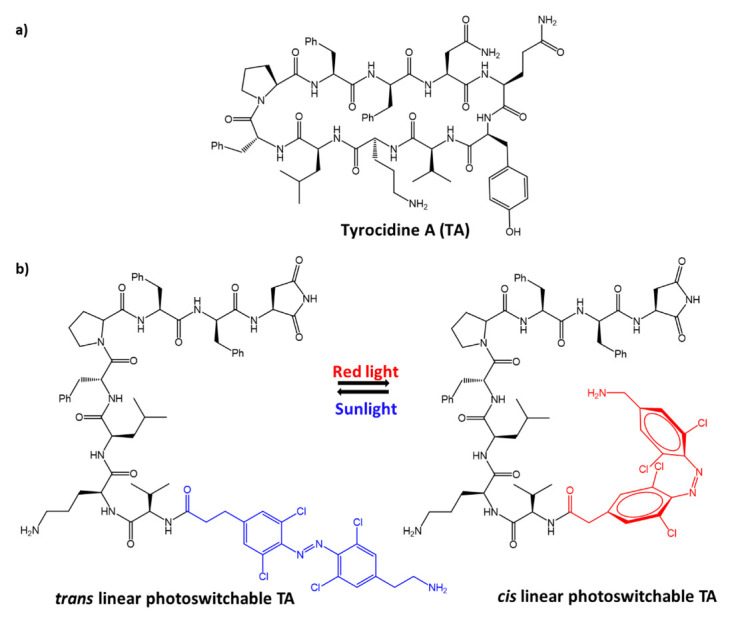
Structures of: (**a**) tyrocidine A; and (**b**) linear photoswitchable analogue of tyrocidine A [85].

**Figure 29 molecules-27-05643-f029:**
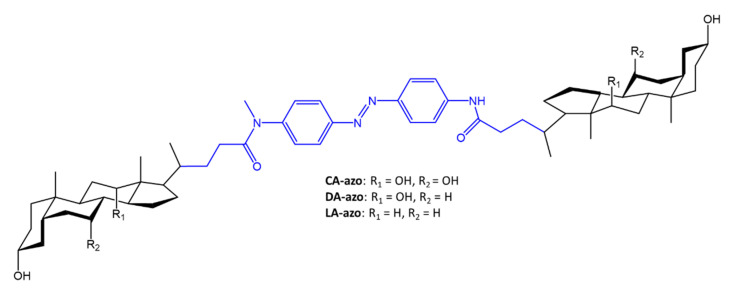
Structure of the azobenzene-linked bile acid dimers [86].

**Figure 30 molecules-27-05643-f030:**
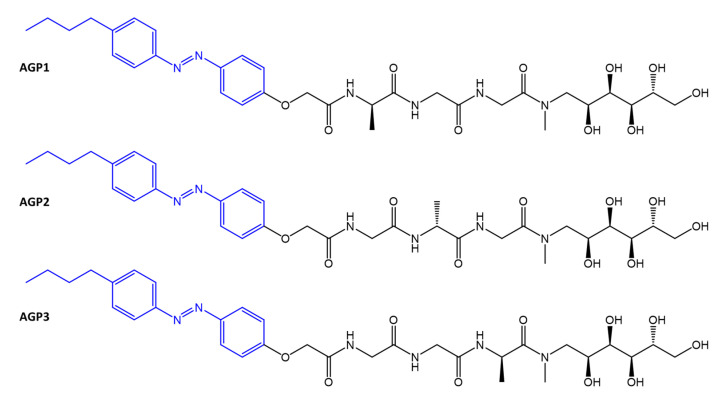
Structures of photoswitchable glycopeptide mimetics [87].

**Figure 31 molecules-27-05643-f031:**
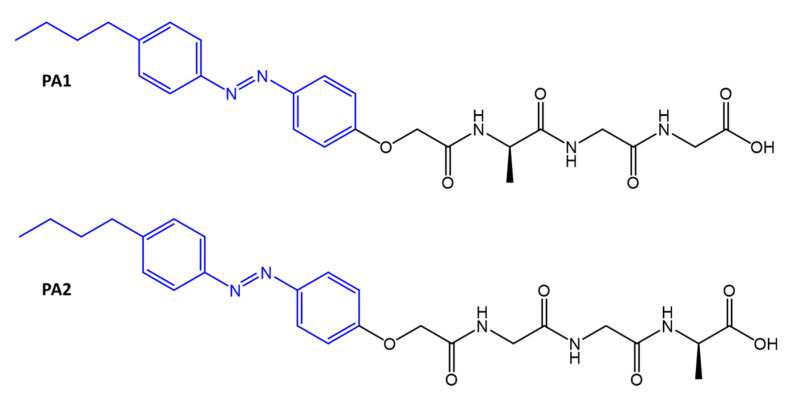
Structures of tripeptide amphiphiles [88].

**Figure 32 molecules-27-05643-f032:**
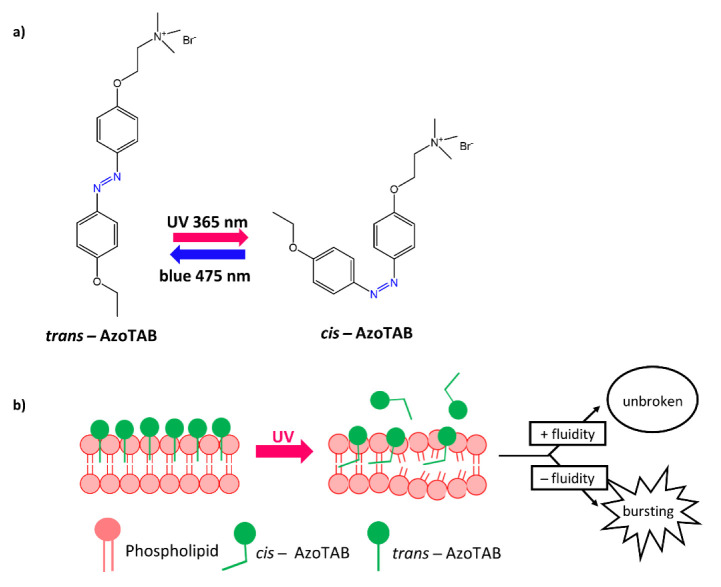
(**a**) structures of *trans*-AzoTAB and *cis*-AzoTAB; (**b**) effect of AzoTAB isomerization on phospholipid membrane [89].

**Figure 33 molecules-27-05643-f033:**
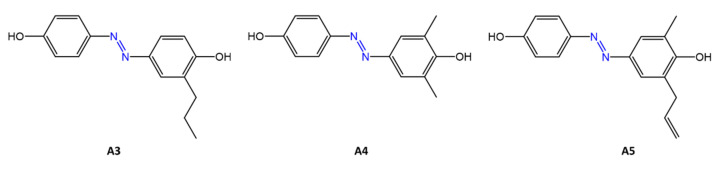
Chemical structures of active azo compounds inserted in PLA films [94].

**Figure 34 molecules-27-05643-f034:**
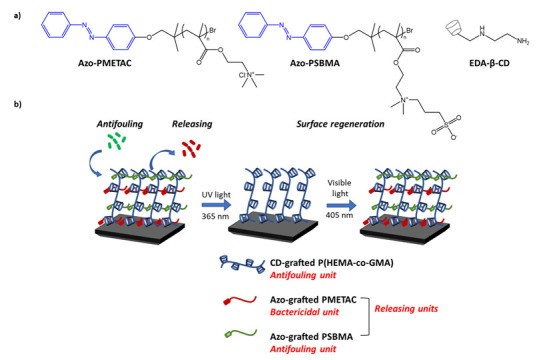
(**a**) chemical structures of components in photoswitchable supramolecular polymer brush; and (**b**) the process of antibacterial surfaces [100].

**Figure 35 molecules-27-05643-f035:**
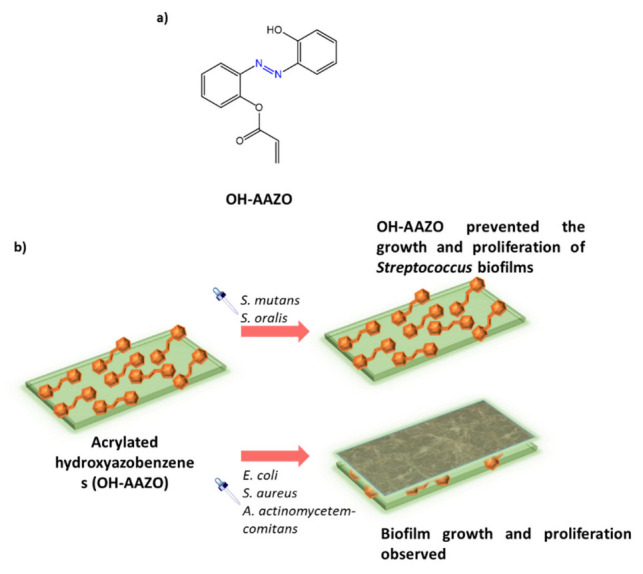
(**a**) chemical structure of OH-AAZO monomer; (**b**) biofilm grown on OH-AAZO coatings and inhibition of Streptococcus biofilms [101].

**Figure 36 molecules-27-05643-f036:**
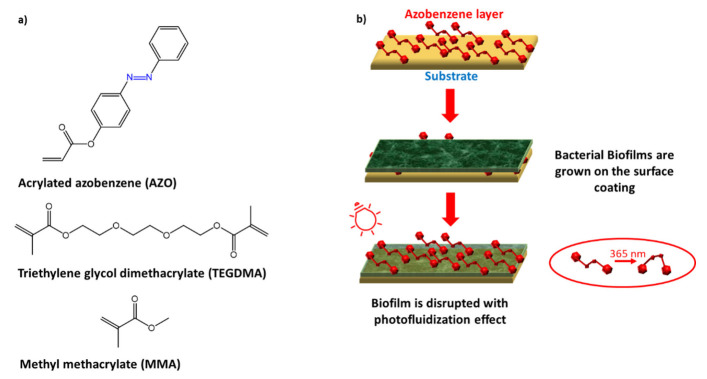
(**a**) chemical structures; (**b**) UV light exposure and biofilm disruption [102].

**Figure 37 molecules-27-05643-f037:**
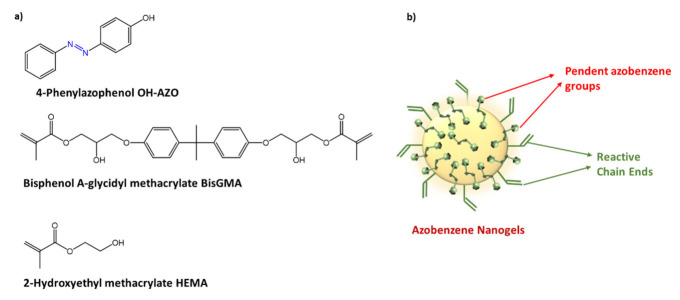
(**a**) chemical structures of nanogels components; (**b**) azobenzene nanogels [103].

**Figure 38 molecules-27-05643-f038:**
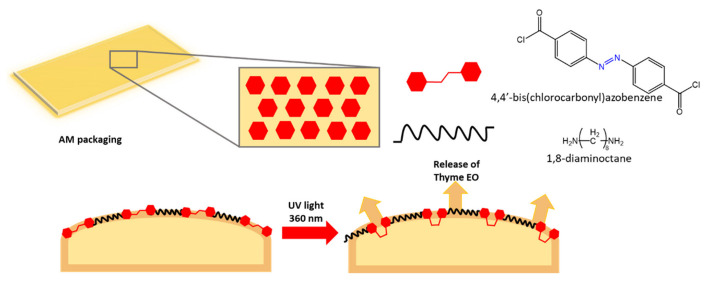
Release mechanism of EO nanocapsules [104].

## Data Availability

Not applicable.

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
