# Peer review of "Azobenzene as Antimicrobial Molecules"

_molecules, 2022, doi:10.3390/molecules27175643_

Round 1

Reviewer 1 Report

The manuscript entitled:"AzoAzobenzenes as antimicrobial molecules" by Simona Concilio and coworkers has been revised. The paper contains a pieze of interesting work in the field of antibiotic resistance by using azomolecules as novel approach to combat bacterial resistance. Different azobencenes derivatives has been reviewed including peptides and organometallics compounds. A great number of schematic figures are wellcome. The paper is well-writting and sound interesting. However, there ate dome aspects that needed improvement before manuscript publication and I feel that minor revision is needed.

The minor points:

- In spite of the great number of figures, the paper lack of systematic overview of the different derivatives and modifications of azobencenes structure and the related effect in important parameters such as MIC or MIC50. Please, include a Table including the chemical modification of azobencenes structure and the related parameters that characterize their antimicrobial effect against bacteria.

-As to the authors claim the wall of bacterial membranes is generally negatively charged, molecules with positively charged groups, such as quaternary ammonium salts, can interact with the bacterial membrane. This assumption is particularly interesting. To reinforce this idea, can the authors put some examples of azobencenes molecules or similar structures in which there is a distinct effect in antibacterial efficience depending on derivative charge? 

- A brief discussion or some lines are needed to include in the introduction section about the use of hybrid materials composed by azobencenes and nanoparticles for drug delivery. 

Author Response

Response to Reviewer 1 Comments

We wish to thank all Reviewers for their thoughtful comments on the original version of our manuscript titled Azobenzenes as antimicrobial molecules.

We carefully considered all comments and suggestions of the Reviewers. Herein, we explain how we revised the manuscript based on these comments and recommendations. As a result, we believe that the manuscript edited in line with all Reviewers' comments has considerably improved and now reached the bar of acceptance.

REVIEWER 1

The minor points:

- In spite of the great number of figures, the paper lack of systematic overview of the different derivatives and modifications of azobencenes structure and the related effect in important parameters such as MIC or MIC50. Please, include a Table including the chemical modification of azobencenes structure and the related parameters that characterize their antimicrobial effect against bacteria.

We thank the reviewer for this suggestion. We prepared a table (Table S1) to highlight the effect of the antibacterial activity of substitutions on the azobenzene ring.

-As to the authors claim the wall of bacterial membranes is generally negatively charged, molecules with positively charged groups, such as quaternary ammonium salts, can interact with the bacterial membrane. This assumption is particularly interesting. To reinforce this idea, can the authors put some examples of azobencenes molecules or similar structures in which there is a distinct effect in antibacterial efficience depending on derivative charge? 

The antibacterial efficacy of azo molecules appears to be enhanced by the polar group, as reported in the cited works of Babamale et al. [48] and Salta et al. [49]. The text has been rephrased accordingly.

- A brief discussion or some lines are needed to include in the introduction section about the use of hybrid materials composed by azobencenes and nanoparticles for drug delivery. 

We modified the text in the introduction following the reviewer's suggestion (lines 69-73 and 86-91).

Reviewer 2 Report

This review article discussed the antimicrobial properties of azobenzene derivatives. In my opinion, there is still room for improvement. Please refer to attached list of comments for revision.

Author Response

Response to Reviewer 2 Comments

We wish to thank all Reviewers for their thoughtful comments on the original version of our manuscript titled Azobenzenes as antimicrobial molecules.

We carefully considered all comments and suggestions of the Reviewers. Herein, we explain how we revised the manuscript based on these comments and recommendations. As a result, we believe that the manuscript edited in line with all Reviewers' comments has considerably improved and now reached the bar of acceptance.

REVIEWER 2

Comments: The review article discussed the antimicrobial properties of azobenzene derivatives. This review will be more comprehensive with the addition of mechanism of action (if any) of these compounds. It can be accepted with revision as mentioned below:

  1. Section 2 of the manuscript discussed the antimicrobial properties of substituted azobenzenes. However, the description of structure-activity relationship between substituents and antimicrobial properties was not thorough. For example, the author did not discuss how the substituents (in terms of types, position and/or number) influenced the antimicrobial properties of compounds reported by Piotto and those in series 4, to name a few.

In a review paper such as this, we summarized the results presented by the authors, and these data are not present in the articles considered. However, we agree with the reviewer that this point is essential and is the subject of a paper we submit. We add some comments in the Introduction. In the supplementary materials section, a table with chemical substitutions on the azobenzene rings and the effect on antibacterial activity. This might be useful for future QSAR works.

  1. In line 179, the authors mentioned that the difference in antimicrobial activity of the cis and trans isomers was due to different permeability ability of the membrane to polar heads. Authors should discuss about the difference with relation to the shape of these isomers.

We added some lines to the manuscript about the shape effect in the conclusion section.

  1. Some of the molecules in Sections 2 and 3 were overlapped in terms of ring substitution, ring modification and association with bioactive moieties. For example, in Section 1, compounds 4(am) (Fig.5) were azobenzene derivatives with ring substitution and ring modification; compounds 8-12 (Fig.6), SBN 1-43 (Fig.10); ASBm, ASBn (Fig.11) were examples of azobenzene associated with bioactive moieties. In section 3, compounds 3a-3k were substituted azobenzenes incorporated with bioactive moiety.

We agree with the reviewer that the text may be confusing. We checked the manuscript, updated the numbering of all molecules, and, most importantly, added a summary table in the supplementary materials of all azobenzene-based molecules reporting the effect on antibacterial activity of chemical substitutions made on the azobenzene scaffold.

  1. This manuscript is a review of the antimicrobial activities of azobenzene derivatives, and hence description of cytotoxic effects of 4a and antitubercular activities of S1-S3 should be excluded. In order to have a comprehensive and systematic discussion, there should be a separate section to discuss about the influence of photoisomerization (cis and trans isomers (linear vs bent)) on antimicrobial properties.

Results concerning cytotoxic and antitubercular properties of compounds 4a and S1-S3 were eliminated.

  1. From section 3 onwards, all figures must be renumbered so to follow the sequence from the previous sections. It is confusing to readers when figures were not consecutively numbered in accordance to their appearance in text.

We agree with the reviewer, and we renumbered all the figures.

  1. Size of all molecular structures shall be standardized, some of which required adjustments. Some of the molecular structures in Figures 13 onwards (section 3 and beyond) were too small, some functional groups in these figures were too close to one another. For example, in Figure 14, the isopropyl and carbonyl groups. In Figure 17, please adjust the bond angles of the side chain of heterocyclic rings and the position of the 3-aminopropyl side chain (points up) in TA. The C-Cl bonds in the cis isomer of TA looked odd. In Figure 23, parentheses were too big.

All the molecular structures have been resized and made the required changes. In the final version of the review, we have already replaced all the figures.

  1. Text in some of the figures were too small (see Figures 6, and some in section 3 onwards). Identities of R1 to R9 were missing in Figure 10; Molecular structures of ligands in Figure 11(Section 4) were missing.

In fig. 10 (section 2) the substituents were added, in fig.22 (section 4) the molecular structure of the ligands was added.

  1. What were GUVs and LUVs (line 483)? The full forms of these acronyms should be provided when they were first mentioned, not in later section (line 654 for GUVs).

We modified the text accordingly.

Round 2

Reviewer 2 Report

I recommend the acceptance and publication of this manuscript.